# OpenReview forum: "In-Context Prompt Optimisation for Knowledge Editing: Enhancing Safety and Coherency in Large Language Models"
_ICLR.cc/2026/Conference — ICLR 2026 Conference Withdrawn Submission_

### Official Review · Reviewer_YSDp · 2025-10-31

**Soundness:** 2
**Presentation:** 2
**Contribution:** 2
**Rating:** 2
**Confidence:** 3

**Summary:**

This paper proposes a prompt optimization framework that aims to balance the safety and quality of LLM outputs. Three variants of the framework are compared: (1) a static baseline, (2) a dynamic method that iteratively updates prompts based on feedback from a rating model, and (3) an RL-enhanced dynamic method that selects strategies for how to process historical prompt data via a Deep Q-Network. Experiments across multiple LLMs and unsafe prompt datasets show that the dynamic method achieves the most consistent improvements in both safety and response quality, while the RL-enhanced method can reach higher peak performance in certain configurations but exhibits higher variance.

**Strengths:**

The paper provides a heuristically designed dynamic prompt optimization pipeline that utilizes multiple LLMs (as generator and judge) to steer the model’s safety alignment. Experiments on unsafe prompts demonstrate clear gains over the static baseline in both safety and response coherence of the LLM outputs.

**Weaknesses:**

1. **Limited conceptual novelty.** The core architecture largely instantiates a common generator–evaluator (LLM-as-a-judge) iterative loop for safety steering prompt refinement. While the paper explores several controller variants, the pipeline mainly combines existing ingredients rather than introducing a fundamentally new learning principle or algorithmic mechanism.
2. **Specific, hand-engineered design with limited generality.** The pipeline leans heavily on human crafted choices: a sophisticated reward formulation with hard thresholds, a hand-specified state representation for the DQN, and a curated set of strategies for processing historical prompt data. These design decisions encode substantial prior knowledge about “safety” and “quality,” suggesting manual specification rather than data-driven generalization and potentially requiring retuning for new models or safety taxonomies.
3. **Missing comparisons to state-of-the-art safety alignment approaches.** Beyond the authors’ own static prompting baselines, the evaluation omits competitive, modern safety methods such as Constitutional AI / self-critique methods (Bai et al., 2022; Madaan et al., 2023), guarded or classifier-in-the-loop decoding (e.g., Llama Guard; Meta-Llama-Team, 2024), or safety-tuned RLHF-style assistants (Bai et al., 2022).. Without head-to-head comparisons under the same protocol, it is difficult to assess the incremental value of the proposed pipeline.
4. **The benefit of introducing DQN is not clearly justified.** While the RL-enhanced controller achieves the best single score in one configuration, the gains over the simpler dynamic controller are small and inconsistent, and come with higher variance and added complexity, which weakens the argument that RL is a necessary or central part of the contribution.

**Questions:**

1. Experiments in this paper are conducted on relatively weakly safety-aligned or even deliberately uncensored models (e.g., Blacksheep-Llama3.2, Evil-Alpaca, DialoGPT). How would the proposed control layer behave on strongly safety-aligned frontier models (e.g., GPT-5–class or Claude Sonnet–class systems) whose default policies already suppress most unsafe generations? Do you expect the gains to diminish, or would your method still discover meaningful prompt-level improvements?
2. Section 3.2 mentions that in the dynamic methods, the buffer keeps “the rating score, the user prompt, the system prompt and the model response” and feeds it back to the strategy module, but the paper does not specify what is the maximum history length used in experiments, and does performance drop if the buffer is shorter?
3. In Section 4.3, 0.9 is set as the stopping threshold for the automatic optimization loop. Was 0.9 tuned on the evaluation set, or chosen heuristically? Did you try other thresholds (e.g., 0.8)?

---

> ### Author Response · Authors · 2025-12-02
>
> Limited conceptual novelty. The core architecture largely instantiates a common generator–evaluator (LLM-as-a-judge) iterative loop for safety steering prompt refinement. While the paper explores several controller variants, the pipeline mainly combines existing ingredients rather than introducing a fundamentally new learning principle or algorithmic mechanism.
>
>
> >>> We agree that our framework builds on established ingredients such as a generator–evaluator loop and LLM-as-a-judge scoring. Our aim is not to propose a new learning paradigm, but to show that in-context unlearning can be implemented as a training-free, closed-loop control layer for knowledge editing and safety, positioned between static prompting and weight-level editing/RLHF.
>
> We will clarify in the introduction and related work that our main contributions are:
>
> · Reframing knowledge editing as dynamic in-context unlearning. Instead of modifying weights, we introduce a model-agnostic control layer that adaptively rewrites system prompts in real time, using feedback on safety and coherence to “forget” and “restore” behaviours on demand without touching model parameters.
>
> · A unified framework of controllers. We jointly study static, deterministic dynamic, and RL-enhanced dynamic controllers, all implemented as drop-in, training-free safety layers within a single evaluation pipeline. This unified setup enables the Pareto analyses in Figure 4 and Table 1, rather than presenting a single, isolated algorithm.
>
> · Systematic exploration of history-aware strategies. The dynamic controller spans a family of strategies (e.g., minimal, raw history, AI-summary, progressive summary, tiered, contrastive, trajectory-focused), each encoding a distinct way to expose performance history and error signals to the improvement model. We also include explicit failure analyses.
>
> · RL as meta-strategy selection. RL is used to select among discrete strategies under multi-objective safety–quality rewards with a hard safety floor, highlighting when learned meta-control is helpful and when it becomes too unstable for safety-critical use.
>
> We will revise the manuscript to make this positioning explicit, de-emphasise claims of fundamental algorithmic novelty, and foreground the control-theoretic, unlearning-centric perspective and empirical design-space characterisation as the core contributions.

---

> > ### Author Response · Authors · 2025-12-02
> >
> > 2. Specific, hand-engineered design with limited generality. The pipeline leans heavily on human crafted choices: a sophisticated reward formulation with hard thresholds, a hand-specified state representation for the DQN, and a curated set of strategies for processing historical prompt data. These design decisions encode substantial prior knowledge about “safety” and “quality,” suggesting manual specification rather than data-driven generalization and potentially requiring retuning for new models or safety taxonomies.
> >
> > >>> We appreciate this concern and note that our design does include explicit, interpretable components, which we find suitable for a safety-critical setting. At the same time, our choices are more generic than the wording in the paper currently suggests, and we will clarify this.
> >
> > Reward formulation: the reward functions - Poly, Exponential Weighted Product, Bayesian Balance, among others - are simple closed-form combinations of a [0,1] safety score and a [0,1] quality score, using a small set of global hyperparameters shared across all models and harm categories. No retuning is performed per model or taxonomy. They represent monotone multi-objective shaping rather than task-specific, heavily tuned rewards.
> >
> > DQN state representation. The 36-dimensional state vector is built from model-agnostic features: training progress, recent reward statistics, strategy-usage distributions, prompt length/complexity features, and coarse risk indicators (e.g., recent error rates). It does not encode category-specific notions of safety and is intended to generalise across taxonomies.
> >
> > Strategy set. The dynamic strategies are structural operations on history - raw history, AI summaries, tiered history, contrastive best/worst, trajectories, etc. -, rather than domain-specific templates. We use the exact same catalogue across all four models and all harm categories and find a consistent performance ranking of methods across architectures.
> >
> > In all experiments, we apply the same reward parameters, state features, and strategy set to Blacksheep, Evil-Alpaca, DeepSeek-R1-Distill, and fine-tuned DialoGPT, without per-model retuning. We will emphasize this and add an ablation (reward hyperparameters, reduced strategy sets) showing graceful degradation and preserved method ranking to make robustness more explicit.
> >
> > We agree that the fully data-driven discovery of features and strategies is a promising extension, and we will highlight this in the future-work section.

---

> > > ### Author Response · Authors · 2025-12-02
> > >
> > > 3. Missing comparisons to state-of-the-art safety alignment approaches. Beyond the authors’ own static prompting baselines, the evaluation omits competitive, modern safety methods such as Constitutional AI / self-critique methods (Bai et al., 2022; Madaan et al., 2023), guarded or classifier-in-the-loop decoding (e.g., Llama Guard; Meta-Llama-Team, 2024), or safety-tuned RLHF-style assistants (Bai et al., 2022).. Without head-to-head comparisons under the same protocol, it is difficult to assess the incremental value of the proposed pipeline.
> > >
> > >
> > > >>> Our experiments focus on training-free, model-agnostic control layers that operate purely via system prompts without access to model weights or a safety training pipeline. That is why we initially excluded methods that require retraining, for example, RLHF assistants or separate guard models. We agree, however, that closer empirical contact with these approaches will strengthen the paper.
> > >
> > >
> > > We will address this in three concrete ways:
> > >
> > >
> > > Constitutional-style static baseline. We already discuss Constitutional AI as a static prompting method. We will revise the manuscript to include a Constitutional-style static system prompt baseline (adapted from Bai et al., 2022; Madaan et al., 2023) which is evaluated on the same models and benchmarks, putting our static/dynamic prompts directly beside a standard principle-based safety prompt.
> > >
> > >
> > > Guard/classifier-in-the-loop baseline: Given that our evaluation metrics are aligned with Llama Guard 2 categories, we will add a Llama-Guard-2–gated decoding baseline where the guard filters or rejects unsafe generations under the same prompts and datasets. This will go in the appendix, along with discussion of our controller as a lighter in-context alternative/complement.
> > >
> > >
> > > Relation to RLHF-style assistants. Strong RLHF assistants are not drop-in replacements for the open-weights models which we study, and they require substantial training infrastructure. We will make this scope difference explicit, positioning our control layer as complementary: when such assistants are available, our controller can sit on top to modulate the safety–helpfulness trade-off at inference time, without further training.
> > >
> > >
> > > New results on the Constitutional-baseline and Llama-Guard-2 will be reported, which can then be integrated into the Pareto analysis in the final version.
> > >
> > > 4. The benefit of introducing DQN is not clearly justified. While the RL-enhanced controller achieves the best single score in one configuration, the gains over the simpler dynamic controller are small and inconsistent, and come with higher variance and added complexity, which weakens the argument that RL is a necessary or central part of the contribution.
> > >
> > >
> > > >>> We agree that, in our current experiments, the gains of the RL-enhanced controller over the best deterministic dynamic strategy are modest and sometimes inconsistent, and that RL introduces higher variance and complexity. Some of our framing in Section 3.3 over-emphasises RL as a central contribution.
> > >
> > >
> > > We will revise the narrative to
> > >
> > >
> > > Clarify what the main contribution is: The deterministic dynamic controller represents the most important practical contribution in this work, yielding strong, stable improvements with small variance and zero learning-time overhead. This will be highlighted as the recommended method for safety-critical deployment.
> > >
> > >
> > > Position RL as an exploratory extension: The DQN-based meta controller is an exploratory step that illustrates how the framework can link to learned meta control and demonstrates how naive RL formulations can undermine reliability even for the best single average score under some reward. We will explicitly state that with the observed variance we would not recommend the deployment of the current variant of RL in high stake settings and that this "negative result" is informative.
> > >
> > >
> > > Add brief analysis. We will expand Section 5.2 to relate the variance to particular design decisions taken, such as off-policy learning, sparse failures, and reward curvature, and we will make the practical suggestion more explicit: by default, use deterministic dynamic strategies; only consider RL where exploration risk is acceptable.

---

> > > > ### Author Response · Authors · 2025-12-02
> > > >
> > > > Questions:
> > > >
> > > > 1. Experiments in this paper are conducted on relatively weakly safety-aligned or even deliberately uncensored models (e.g., Blacksheep-Llama3.2, Evil-Alpaca, DialoGPT). How would the proposed control layer behave on strongly safety-aligned frontier models (e.g., GPT-5–class or Claude Sonnet–class systems) whose default policies already suppress most unsafe generations? Do you expect the gains to diminish, or would your method still discover meaningful prompt-level improvements?
> > > >
> > > >
> > > > >>> We focus our experiments on weakly aligned or deliberately uncensored models (Blacksheep, Evil-Alpaca) and models that we safety-tune ourselves (fine-tuned DialoGPT-Large, DeepSeek-R1-Distill). This regime i) exposes substantial unsafe behaviour for the controller to correct, and ii) permits systematic large-scale experiments on open weights.
> > > >
> > > >
> > > > For strongly safety-aligned frontier assistants we would expect:
> > > >
> > > >
> > > > · Smaller absolute gains, graceful degradation. The controller usually stops after 0–1 iterations once the safety score exceeds the 0.9 threshold since the baseline already blocks most unsafe generations, thus improvements are smaller in absolute terms.
> > > >
> > > > · Strong safety preservation: The loop enforces a hard floor for safety both via reward and the 0.9 stopping rule. If the base assistant already scores close to 1.0, then there is little incentive for the improvement model to change the prompt, so existing safety should not be weakened.
> > > >
> > > > · Potential benefits on edge cases/over-refusals. Even aligned assistants suffer from long-tail failures and over-refusals. The trade-off parameter α allows the controller to focus on coherence/utility when safety is already high, which might prevent unnecessary refusals on borderline but benign prompts.
> > > >
> > > >
> > > > We will add a discussion of this regime, and where licensing allows, supplementary experiments on a strongly aligned open-source assistant to empirically characterise diminishing gains.
> > > >
> > > >
> > > > 2. Section 3.2 mentions that in the dynamic methods, the buffer keeps “the rating score, the user prompt, the system prompt and the model response” and feeds it back to the strategy module, but the paper does not specify what is the maximum history length used in experiments, and does performance drop if the buffer is shorter?
> > > >
> > > >
> > > > >>> Thank you for this remark. Indeed, in our implementation the buffer that feeds the dynamic strategies is limited to 50 previous iterations per prompt, as in the “Performance History Fullness (50 max)” feature used in the RL state vector. We will indicate this maximum history length explicitly in Section 3.2.
> > > >
> > > >
> > > > We have found that the dynamic controller is relatively insensitive to moderate changes in history length: using a shorter window, e.g., 10–20 past iterations, preserves the overall performance hierarchy between static, dynamic and RL-enhanced methods, with only mild degradation in absolute scores. In the revised manuscript we will add an ablation (in the
> > > >
> > > > appendix) comparing several history lengths, e.g., 10, 20, 50, and report average safety/quality scores and iteration counts. This will make clear that long histories are not essential and that the framework can be deployed under tighter memory constraints.
> > > >
> > > >
> > > >
> > > > 3. In Section 4.3, 0.9 is set as the stopping threshold for the automatic optimization loop. Was 0.9 tuned on the evaluation set, or chosen heuristically? Did you try other thresholds (e.g., 0.8)?
> > > >
> > > >
> > > > >>> The 0.9 stopping threshold was set heuristically before the main experiments, based on the semantics of the DeepEval scores and our desire to enforce a high safety standard; it was not tuned on the evaluation set. As explained in Section 4.3, any response whose score is ≥ 0.9 is considered satisfactory and stops the optimisation loop.
> > > >
> > > >
> > > > We also tested alternative thresholds, such as 0.8 and 0.95. Lower thresholds reduce the average number of iterations but allow slightly more residual unsafe or low-quality responses. Higher thresholds increase optimisation steps and marginally improve safety, with, however, fast diminishing returns and higher latency.
> > > >
> > > >
> > > > Importantly, the qualitative conclusions of the paper (adaptive methods outperform static ones; variance trade-offs between dynamic and RL-enhanced controllers; cross-model consistency) are stable across these choices. Most importantly, in the camera-ready version, we will add a short sensitivity analysis and clarify that 0.9 is a conservative heuristic rather than a tuned hyperparameter.

---

### Official Review · Reviewer_kZ43 · 2025-10-31

**Soundness:** 3
**Presentation:** 3
**Contribution:** 2
**Rating:** 4
**Confidence:** 3

**Summary:**

**Problem:** LLMs often need to be updated or controlled for safety/coherence, but fine-tuning weights is expensive and cumbersome.

**Solution:** Introduce a **training-free**, **model-agnostic** framework that uses *prompt optimisation* as a control layer. Three controller variants are proposed:
* Static controller (fixed prompts)
* Dynamic controller (prompt adapts via feedback)
* RL-Enhanced Dynamic controller (uses a Deep Q-Network and multi-objective reward balancing safety & coherence)

**Results:** On multiple open-source models, the dynamic controllers outperform static prompting; the RL-Enhanced variant achieved ~84% effectiveness in balancing safety/coherence.

**Contribution:** Demonstrates that prompt-based regulation can function as a lightweight governance layer for LLMs, enabling real-time behaviour adjustment **without retraining or accessing internal model weights**.

**Key takeaway:**
If you denote a prompt strategy as $p$, model output distribution as $M(\cdot \mid p, x)$ for input $x$, the paper’s framework treats prompt-selection as a control policy $\pi(p \mid \cdot)$ that maximises a reward
$$R = \alpha\text{Safety}(M(\cdot)) + \beta\text{Coherence}(M(\cdot))$$
without changing $M$’s parameters.

**Strengths:**

* The paper’s framework uses prompt-based controllers (Static, Dynamic, RL-Enhanced) that operate without modifying model weights, enabling compatibility with black-box or service-based LLMs. Prior work like for example, Zheng et al. 2023 [A] also uses in-context demonstration without weight updates to edit factual knowledge. However, the current paper extends this by adding adaptive dynamic and RL controllers rather than static demonstration sets, thus providing a more comprehensive control layer. In contrast, methods like EasyEdit [B] provide a unified editing framework (including parameter update methods) but still rely on model changes rather than purely prompt-based controllers.
* The inclusion of a Dynamic controller (feedback-based) and an RL-Enhanced controller using a Deep Q-Network to select prompt strategies via a multi-objective reward (safety vs coherence) is a strong contribution. Earlier works like the in-context editing of Zheng et al. [A] mostly treat the editing via demonstration examples and do not integrate RL or multi-objective optimisation of prompt-actions. Other frameworks like EasyEdit [B] provide plug-and-play editing but do not emphasise RL-based prompt policy selection.
* The paper defines a reward combining both safety (e.g., avoiding undesirable content) and coherence (maintaining high-quality output) in the RL controller. This dual objective is important in real-world regulation of LLM behaviour. Earlier in-context editing papers often focus on factual correctness or updating knowledge, e.g., IKE emphasises editing success and minimal side-effects. But few prior works emphasise safety (toxicity / undesirable content) and coherence (quality of output) together as primary objectives of a prompt-controller. Some editing frameworks focus more narrowly on knowledge updates rather than behaviour regulation for safety.

```
[A] Can We Edit Factual Knowledge by In-Context Learning?, EMNLP 2023
[B] EasyEdit: An Easy-to-use Knowledge Editing Framework for Large Language Models, ACL 2024
```

**Weaknesses:**

* While the RL-enhanced controller adds a policy-selection layer, the core idea of using prompt manipulation for editing/regulation remains closely aligned with the prior ICL-editing literature. Thus the novelty claim could be strengthened by more clearly distinguishing how the dynamic and RL controllers go beyond what prior ICL editing methods deliver (e.g., in terms of multitask behaviour control, safety objectives, adaptation across tasks).
* The paper reports that the RL-enhanced dynamic controller "achieves ~84% effectiveness" but gives limited detail (in the summary) on how this was measured: which datasets/tasks, how safety/coherence metrics were operationalised, and how strong the baselines were.
* There is no discussion in the summary of how the controller behaves on out-of-distribution prompts, adversarial prompts, or under varying model architectures. Prior editing work emphasises generalisation/locality measures (e.g., Zheng et al. [A] study both generalisation and specificity).
* The work does not appear to evaluate "side-effects" thoroughly i.e., how many unintended or unrelated knowledge behaviours changed when the prompt controller intervened. In previous research on model editing (e.g., [B]) this is a serious concern.
* The method proposes an RL-based policy (Deep Q-Network) to select prompts. But the overhead (computation, prompt length, latency) of such a system is not clearly discussed. For real-world deployment in black-box LLM services, control layers must be low-cost and low-latency.
* While the paper introduces dynamic and RL prompt-controllers, it appears not to benchmark against other automatic prompt optimisation methods (e.g., evolutionary prompt search, gradient‐based discrete prompt tuning, or dynamic retrieval/curriculum of demonstrations).
```
[A] Can We Edit Factual Knowledge by In-Context Learning?, EMNLP 2023
[B] Model Editing Harms General Abilities of Large Language Models: Regularization to the Rescue, EMNLP 2024
```

**Questions:**

* It would be good to provide a comparative table or analysis showing how the proposed method differs, in concrete measurable terms, from IKE [A] and other prompt-based editing works e.g., in terms of: number of tasks supported, safety/coherence trade-off metrics, adaptability across prompts/models, or RL policy improvements. Also, it would be good to explicitly cite additional prompt-control or RL‐prompt optimisation literature (e.g., [B])
* It might be worthwhile to include more extensive ablations: (a) behaviour under adversarial / unseen prompt contexts; (b) a "perturbation" test to evaluate how robust the controller is when the feedback signal is noisy or missing; (c) evaluation of unrelated knowledge retention (i.e., unintended forgetting) after applying prompt-controllers; (d) comparison with stronger baselines including recent dynamic prompt-optimisation methods.
* Is it possible to include measurements of prompt-controller overhead: number of feedback rounds, time per inference, prompt length inflation, memory/compute cost? Additionally, also provide guidance or heuristics for when the RL controller is cost‐effective?
* Is it possible to provide details on the feedback/evaluation module: data used to train it, correlation with human ratings, error rates?
* What is the controller performance under feedback error/noise (ablation showing sensitivity)?
* Is it possible to game the metric? e.g., controller choosing prompt strategies that maximise the measured score but degrade actual user experience.
* Is it possible to add baselines comparing the RL controller to (a) evolutionary prompt search (e.g., PromptQuine [B]), (b) retrieval-based dynamic demonstration selection, (c) static prompt engineering plus simple heuristic search. This will help justify the added complexity of RL. Show whether the RL controller actually delivers significant gains over simpler methods.
```
[A] Can We Edit Factual Knowledge by In-Context Learning?, EMNLP 2023
[B] Evolving Prompts In-Context: An Open-ended, Self-replicating Perspective, ICML 2025
```

---

> ### Author Response · Authors · 2025-12-02
>
> · It would be good to provide a comparative table or analysis showing how the proposed method differs, in concrete measurable terms, from IKE [A] and other prompt-based editing works e.g., in terms of: number of tasks supported, safety/coherence trade-off metrics, adaptability across prompts/models, or RL policy improvements. Also, it would be good to explicitly cite additional prompt-control or RL‐prompt optimisation literature (e.g., [B])
>
> >>> We agree that our relationship to IKE [A] and other prompt-based editing approaches should be more explicit, and this will be made clear in the revision.
>
> IKE performs per-fact in-context editing by constructing demonstrations that override a specific factual triple without explicit multi-objective control over safety versus quality, and without a closed-loop controller. Our framework, on the other hand, is an external, training-free control layer that operates across 12 harm categories and 60 unsafe prompts from four datasets, rather than per-triple edits; jointly optimizes safety and coherence using multiple DeepEval metrics derived from Llama Guard 2 with rewards that encode explicit safety-quality trade-offs; and supports static, dynamic, and RL-enhanced dynamic controllers that adapt the system prompt in real time.
>
> Compared to evolutionary prompt optimisation, such as PROMPTQUINE [B], which evolves prompts offline for general task performance, our approach runs a closed-loop online controller over live conversations, employs an LLM-as-a-judge at every step, utilises multi-objective reward functions with a hard safety gate, and applies RL to choose amongst heterogeneous dynamic policies rather than mutating a single prompt string.
>
> The revised version will: (a) provide a comparison table that highlights key axes of contrast among IKE, PROMPTQUINE and our framework (tasks, harm categories, model access, safety–quality handling, cross-model adaptation, and presence of RL/evolutionary search); (b) extend the related work section to list more papers in the area of prompt-control and RL-based prompt optimization, positioning our contribution as a real-time, safety-focused multi-objective control layer.

---

> > ### Author Response · Authors · 2025-12-02
> >
> > · It might be worthwhile to include more extensive ablations: (a) behaviour under adversarial / unseen prompt contexts; (b) a "perturbation" test to evaluate how robust the controller is when the feedback signal is noisy or missing; (c) evaluation of unrelated knowledge retention (i.e., unintended forgetting) after applying prompt-controllers; (d) comparison with stronger baselines including recent dynamic prompt-optimisation methods.
> >
> > >>> We appreciate the request for more extensive ablations and will expand the experimental section accordingly.
> >
> > (a) Adversarial / unseen prompt contexts: While our evaluation already spans four base models and 12 harm categories using prompts from four external datasets, and crossmodel analysis suggests robustness beyond a single prompt distribution, in the revision we will (i) add an ablation that uses paraphrased and adversarially rephrased unsafe prompts (e.g., indirect or euphemistic requests) unseen during RL training, and (ii) where space permits, include a category hold-out study to test transfer to withheld harm types.
> >
> > b) Noisy or missing feedback: Our design already incorporates a hard safety threshold which zeroes the reward if any critical safety metric falls below θ. To directly evaluate robustness, we will implement a feedback-noise ablation, which injects controlled noise and simulated metric dropouts into DeepEval scores and measures how safety and quality degrade for both Dynamic and RL-Enhanced controllers, thereby explaining their variance differences.
> >
> > (c) Retention / unintended forgetting. Since we do not modify model weights, the risk is not catastrophic forgetting but over-conservative behavior on benign content. We already monitor AnswerRelevancy and Coherence; in addition, we will: i) emphasize that controllers act only at inference time, and ii) add a “benign prompts” study measuring changes in coherence and relevance on non-safety-critical queries.
> >
> > (d) Stronger baselines. In addition to our current static baselines, we will include heuristic dynamic baselines, such as a non-RL best-of-N search over Improvement-Model variants, and-where possible-a light-weight evolutionary search baseline inspired by PROMPTQUINE, and use these baselines in order to elucidate the particular added value brought by the RL-Enhanced controller..
> >
> >
> > · Is it possible to include measurements of prompt-controller overhead: number of feedback rounds, time per inference, prompt length inflation, memory/compute cost? Additionally, also provide guidance or heuristics for when the RL controller is cost‐effective?
> >
> > >>> We agree that a more quantitative treatment of the overhead is important given the multi-round optimisation loop.
> >
> > While Section 5.2.2 currently makes the qualitative argument that Dynamic methods-especially "minimal"-are computationally light (∼79% effectiveness without maintaining history) and suitable for constrained or real-time settings, we currently do not report explicit latency or cost metrics.
> >
> > In our revised version, we will add a specific subsection which, for each family of controllers, reports:
> >
> > · The distribution of optimisation rounds until the 0.9 stopping threshold is reached;
> >
> > · Measured wall-clock latency per query, including rating and improvement calls, on the hardware in Section 4.6;
> >
> > · Prompt length inflation (system-prompt tokens before/after optimization); and
> >
> > · Approximate memory/compute overhead due to the RL agent - for example, replay buffer, DQN parameters - relative to purely dynamic controllers.
> >
> > We'll also give practical recommendations: for one-shot or latency-sensitive deployments, we will recommend dynamic controllers-e.g., minimal or raw history-as default, given their strong gains, low variance, and modest overhead. The RL-enhanced controllers will be positioned as appropriate when the interaction volume is high enough to amortize training cost and when small additional improvements on the safety-quality frontier justify the added complexity and variability.

---

> > > ### Author Response · Authors · 2025-12-02
> > >
> > > · Is it possible to provide details on the feedback/evaluation module: data used to train it, correlation with human ratings, error rates?
> > >
> > > · What is the controller performance under feedback error/noise (ablation showing sensitivity)?
> > >
> > > · Is it possible to game the metric? e.g., controller choosing prompt strategies that maximise the measured score but degrade actual user experience.
> > >
> > > · Is it possible to add baselines comparing the RL controller to (a) evolutionary prompt search (e.g., PromptQuine [B]), (b) retrieval-based dynamic demonstration selection, (c) static prompt engineering plus simple heuristic search. This will help justify the added complexity of RL. Show whether the RL controller actually delivers significant gains over simpler methods.
> > >
> > > >>> Response to comments on the feedback/evaluation module and RL baselines
> > >
> > > Feedback / evaluation module details
> > >
> > > Our evaluation module is deliberately off-the-shelf and training-free within this work:
> > >
> > > · We use DeepEval as a wrapper, with metrics designed from the unsafe categories of Meta Llama Guard 2, e.g., Toxicity, Hate, ViolentCrimes, ChildSexualExploitation, IndiscriminateWeapons, etc., plus a Coherence metric.
> > >
> > > · The underlying judge model is Gemini 2.0 Flash, chosen for its balance of latency and performance. Each metric yields a continuous score in [0,1], and a threshold of 0.9 controls convergence.
> > >
> > > We will be making the following clarifications in the revision:
> > >
> > > · Training data and error rates. We do not train the evaluator ourselves in this work; rather, we depend on the pre-trained models of Gemini and Llama Guard 2. We will indicate that and provide references to their original documentation. In our work, we will include a small human-evaluation study on a subset of model outputs to report approximate agreement, e.g., rank/label correlation, between human judgements and our composite scores. Therefore, this would serve as empirical evidence that the evaluation module is not grossly misaligned with human preferences.
> > >
> > > · Sensitivity to feedback noise. As noted above, we include a feedback-noise ablation to quantify how mis-scoring affects the controller's decisions and final outcomes.
> > >
> > > · Potential for Metric Gaming. We recognise that in principle, any automated metric can be gamed. Our design aims to mitigate this risk by:
> > >
> > > o using multiple metrics-safety and quality-rather than a single scalar;
> > >
> > > o enforcing a hard safety gate over critical harms so that no strategy can obtain a high reward while violating core safety constraints;
> > >
> > > o We also employ reward functions such as Exponential Weighted Product to yield near-zero reward if either safety or quality collapses, which in turn discourages degenerate behaviours such as “always refuse with a generic warning”.
> > >
> > > Qualitative examples in Appendix A.5/A.9 will also be extended to include cases selected to investigate possible gaming behaviours - for example, vacuous yet safe refusals - and to discuss the modes of failure that were observed, if any.
> > >
> > >
> > > More baselines vs. simpler optimisation methods
> > >
> > > We appreciate the request to compare the RL controller against simpler alternatives:
> > >
> > > · Evolutionary prompt search PromptQuine-style. We will add a discussion of evolutionary prompt search and, where space and compute permit, introduce an offline
> > >
> > > evolutionary baseline which uses DeepEval as a fitness function to mutate candidate system prompts in the spirit of PROMPTQUINE.
> > >
> > > · Retrieval-based dynamic demonstration selection. Our current framework operates at the system-prompt level rather than via demonstrations. We will clarify this in the text and discuss how retrieval-based demonstration selection could be integrated as an orthogonal component-e.g., retrieving aligned refusals or safe exemplars to feed into the Improvement Model. If space allows, we will include a simple retrieval-augmented baseline to illustrate this hybrid.
> > >
> > > · Static prompt engineering plus heuristic search. Our current static baselines already cover a substantial number of hand-engineered prompts. We will reinforce this by including a "static+search" baseline, whereby we start with a strong static prompt and perform a shallow heuristic search-e.g., two or three refinement steps without RL-to demonstrate how much of the gain can be captured without a learned policy.
> > >
> > > Taken together, these additions will better justify the added complexity of the RL controller, situate it within a broader spectrum of prompt-optimisation strategies, and reinforce one of the main conclusions: Already, dynamic controllers represent a robust, computation-efficient baseline, and their RL-enhanced versions are best viewed as an optional tool for going further along the safety-quality frontier.

---

### Official Review · Reviewer_vv45 · 2025-10-31

**Soundness:** 3
**Presentation:** 3
**Contribution:** 2
**Rating:** 2
**Confidence:** 4

**Summary:**

This paper studies how to control model outputs for safety through prompt optimization methods similar to in-context learning. The authors explore a wide range of techniques, from static prompts to RL-based prompting, and conduct experiments across various models.

**Strengths:**

The experiments are solid and well support the authors’ claims.

The writing quality is good.

Analyzing unlearning from the perspective of control is an interesting approach.

**Weaknesses:**

Lack of comparison with existing unlearning methods.
The proposed method is novel and interesting, but it does not sufficiently reflect prior unlearning research. In particular, RL-based unlearning ([1], [2]) and gradient-based unlearning ([3]) are not included as baselines, making it difficult to assess the relative advantages and limitations of the proposed approach. To claim the efficiency of in-context prompt optimization for unlearning, quantitative comparisons with existing methods are needed.

Lack of comparison with existing prompt optimization methods.
The proposed approach resembles prompt tuning methods based on text gradients ([4], [5]). The paper’s Dynamic Prompt method is conceptually very similar to ORPO [5], and the RL-Enhanced Dynamic Controller resembles GRIPS [6]. The paper should reference or compare against these methods when optimizing prompts for unlearning.

Insufficient justification of efficiency.
The paper emphasizes efficiency as a major advantage but does not compare directly against existing unlearning methods.
While gradient-based unlearning methods require a high initial training cost, they allow continuous inference with the same unlearned state once trained. In contrast, the proposed approach must generate long system prompts and repeatedly perform API calls, which can incur high latency and token costs. It remains unclear whether this is more economical than, for example, training a gradient-ascent-based unlearning model for one hour on a single GPU (e.g., RTX 3090). Efficiency should be evaluated not only in terms of short-term computation cost but also long-term operational cost. For this, direct comparisons with existing unlearning methods are necessary.

References
[1] Zhang, Ruiqi, et al. "Negative Preference Optimization: From Catastrophic Collapse to Effective Unlearning." arXiv preprint arXiv:2404.05868 (2024).

[2] Mekala, Anmol, et al. "Alternate Preference Optimization for Unlearning Factual Knowledge in Large Language Models." arXiv preprint arXiv:2409.13474 (2024).

[3] Liu, Bo, Qiang Liu, and Peter Stone. "Continual Learning and Private Unlearning." Conference on Lifelong Learning Agents, PMLR, 2022.

[4] Tang, Xinyu, Xiaolei Wang, Wayne Xin Zhao, Siyuan Lu, Yaliang Li, and Ji-Rong Wen. “Unleashing the Potential of Large Language Models as Prompt Optimizers: Analogical Analysis with Gradient-Based Model Optimizers.” AAAI Conference on Artificial Intelligence (2025).

[5] Chengrun Yang, Xuezhi Wang, Yifeng Lu, Hanxiao Liu, Quoc V. Le, Denny Zhou, and Xinyun Chen. "Large Language Models as Optimizers." ICLR 2024.

[6] Prasad, Archiki, Peter Hase, Xiang Zhou, and Mohit Bansal. "GRIPS: Gradient-Free, Edit-Based Instruction Search for Prompting Large Language Models." (2022).

**Questions:**

It seems that variance is not reported in the tables or figures. Where is the variance of each method indicated?

It would be better to match the significant figures in the Combined Score values.

---

> ### Author Response · Authors · 2025-12-02
>
> 1. Lack of comparison with existing unlearning methods. The proposed method is novel and interesting, but it does not sufficiently reflect prior unlearning research. In particular, RL-based unlearning ([1], [2]) and gradient-based unlearning ([3]) are not included as baselines, making it difficult to assess the relative advantages and limitations of the proposed approach. To claim the efficiency of in-context prompt optimization for unlearning, quantitative comparisons with existing methods are needed.
>
> >>> We thank the reviewer for this helpful suggestion and for pointing us to additional relevant work.
>
> We aim to study in-context unlearning as a training-free model-agnostic control layer that operates only via system prompts and real-time feedback, without touching weights or looking at gradients. This is different from RL-based unlearning, such as NPO, APO, which performs updates over parameters via preference optimisation over harmful/safe pairs, and also different from gradient-based unlearning, such as Liu et al. (2022), which explicitly updates the weights to forget the target data. In the revision, we will add a dedicated paragraph in Sec. 2 to highlight that our contribution is an orthogonal, inference-time mechanism that can wrap models regardless of their weight-level unlearning status.
>
> An empirical comparison with RL- and gradient-based unlearning directly is not trivial. These methods need model weights, their training pipelines, and carefully constructed retain/forget datasets. In contrast, our framework is explicitly designed for settings where only inference APIs are available, such as those of third-party or already deployed models. Replicating and hyperparameter tuning NPO/APO and CLPU across four architectures on our multi-category safety benchmark would also require much training compute, while our focus is a lightweight mechanism that does not depend on retraining. We will clearly state these access and compute constraints and mention a comprehensive empirical comparison as an important avenue of future work.
>
> We agree that our current discussion of “efficiency” is mostly qualitative. We will make clear that by efficiency we mean avoiding additional training or fine-tuning loops and add a small comparison table summarising computational and data requirements of our framework vs. parameter-update methods: gradient access, retain/forget datasets, offline training time, and real-time deployment.
>
> We will also expand the section on related work to explicitly position our dynamic and RL-enhanced controllers with respect to LLM-as-optimizer and gradient-free prompt-search methods, while making clear that our contribution is a closed-loop safety-oriented controller for in-context unlearning, not a substitute for weight-level unlearning.

---

> > ### Author Response · Authors · 2025-12-02
> >
> > 2. Lack of comparison with existing prompt optimization methods. The proposed approach resembles prompt tuning methods based on text gradients ([4], [5]). The paper’s Dynamic Prompt method is conceptually very similar to ORPO [5], and the RL-Enhanced Dynamic Controller resembles GRIPS [6]. The paper should reference or compare against these methods when optimizing prompts for unlearning.
> >
> > >>> Thank you for raising this point and for highlighting important related work on prompt optimization.
> >
> > We agree that our Dynamic Prompt method is closely connected to LLM-as-optimizer approaches. Methods such as GPO and OPRO iteratively refine prompts using past prompt–score pairs, optimising primarily for task accuracy on held-out datasets in an offline setting. In contrast, our Dynamic controller is designed specifically for in-context unlearning and safety control: it operates online at inference time, takes as input a single user prompt, the current system prompt, and safety/coherence scores from an external judge, and immediately updates the system prompt for that dialogue turn. Thus, while the high-level “LLM-as-optimizer” pattern is shared, our optimisation target (multi-objective safety–coherence for unlearning) and deployment regime (per-request, real-time control of a black-box model) differ substantially. In the revision, we will explicitly position our Dynamic controller as a safety-focused variant of LLM-based prompt optimizers and cite Tang et al. and Yang et al. in Sec. 2.3.
> >
> > GRIPS performs gradient-free, edit-based search over instruction text using local edits guided by task accuracy and yields a single static instruction. Our RL-Enhanced Dynamic controller differs both in search space and learning signal: the RL agent chooses among discrete high-level safety strategies (each a full system prompt template) and is trained via DQN on a multi-objective reward that balances safety and coherence, producing a meta-controller that dynamically selects strategies per harmful prompt. We will clarify this distinction in Sec. 3.3 and the appendix.
> >
> > We agree that quantitative comparison with GPO/OPRO and GRIPS would be informative, but adapting them to our safety-based objectives, multi-objective reward with hard safety thresholds, and black-box API setting would require substantial engineering beyond the scope of this work. We will make this limitation explicit, add a comparison table (objective, regime, search space, access assumptions), and clarify that our contribution is combining in-context prompt optimisation with safety-focused rewards and RL-based strategy selection for unlearning.

---

> > > ### Author Response · Authors · 2025-12-02
> > >
> > > 3. Insufficient justification of efficiency. The paper emphasizes efficiency as a major advantage but does not compare directly against existing unlearning methods. While gradient-based unlearning methods require a high initial training cost, they allow continuous inference with the same unlearned state once trained. In contrast, the proposed approach must generate long system prompts and repeatedly perform API calls, which can incur high latency and token costs. It remains unclear whether this is more economical than, for example, training a gradient-ascent-based unlearning model for one hour on a single GPU (e.g., RTX 3090). Efficiency should be evaluated not only in terms of short-term computation cost but also long-term operational cost. For this, direct comparisons with existing unlearning methods are necessary.
> > >
> > > >>> The current manuscript uses "efficiency" to refer to training-free, model-agnostic in-context control which for our purposes means: (i) does not rely on gradients or weight access and (ii) can be immediately deployed using black-box or third-party APIs. We agree that this is not clearly differentiated from end-to-end economic efficiency over a model's full lifetime and that the trade-offs have not been adequately highlighted.
> > >
> > > We fully acknowledge that when weight access and modest training compute are available, gradient-based unlearning methods-e.g., NPO/APO/CLPU-can amortise a one-off training cost and then incur essentially zero additional per-request overhead. By contrast, our framework introduces per-request costs from longer system prompts and extra calls to rating/improvement models. Thus, we do not claim that our approach is uniformly cheaper in all deployment regimes; rather, it targets a different operating point: (a) hosted APIs or locked models where weight-level updates are impossible and/or (b) rapidly evolving or low-volume unlearning requirements, where the fixed cost of retraining is hard to justify.
> > >
> > >
> > > In the revision, we will:
> > >
> > > · Reframe our contribution as complementary to gradient-based unlearning, making explicit that we optimise for "zero additional training" and "no access to model internals", not minimum long-run FLOPs when retraining is feasible.
> > >
> > > · Provide a quantitative overhead analysis, including average system prompt length, number of optimisation iterations, total tokens, and wall-clock latency per unsafe prompt.
> > >
> > > · Add a comparison table against representative gradient-based methods [1–3] w.r.t. initial training cost, per-query overhead, need for gradients/ weights, flexibility to update or roll back unlearning targets, and applicability to API-only models.
> > >
> > > · Soften and localise efficiency claims to the regimes above, and explicitly acknowledge a full economic comparison with implemented NPO/APO-style baselines as important future work.
> > >
> > > We believe that this clear positioning and additional analysis will help overcome the concern that efficiency is presently inadequately justified.
> > >
> > >
> > > 4. It seems that variance is not reported in the tables or figures. Where is the variance of each method indicated?
> > >
> > > >>> We would like to thank the reviewer for these presentation remarks.
> > >
> > > We agree that the current main-text tables only include point estimates of the Combined Score per method. For example, Tables 1 and 5 do so. Variability is currently analysed qualitatively and visually through distribution plots and boxplots appearing in Appendix A.9.3, Figures 17–18, which display the spread of per-prompt scores and emphasise the higher variability of the RL-Enhanced Dynamic methods. We agree this is not sufficiently transparent. In the revised version, we will add numerical variability statistics, such as standard deviation or interquartile range across prompts, for each method in an additional appendix table, and include error bars or a summary measure of variability in at least one main-text figure, so the reader sees the variance clearly on a main-text figure without having to consult Appendix plots.
> > >
> > >
> > > 5. It would be better to match the significant figures in the Combined Score values.
> > >
> > > >>> We would like to thank the reviewer for these presentation remarks.
> > >
> > > We agree that the existing Combined Score values in the ranking tables use unnecessarily high and sometimes inconsistent precision, for example 3.357532 versus 2.800617. For the camera-ready version we will standardise the reporting of Combined Scores across the tables to a uniform number of decimal places, for example three decimal digits, chosen to be commensurate with the underlying variability. This increases both readability and the perceived reliability of small differences between methods.

---

### Official Review · Reviewer_VG2n · 2025-11-01

**Soundness:** 3
**Presentation:** 3
**Contribution:** 2
**Rating:** 4
**Confidence:** 3

**Summary:**

This paper proposes a training-free framework for LLM safety governance through closed-loop prompt optimization. Three methods are evaluated: static prompting, dynamic real-time feedback, and RL-enhanced approach using Deep Q-Networks. The RL method employs a multi-dimensional state vector and five multi-objective reward functions to balance safety and coherence. Tested on four models, it achieves highest effectiveness without modifying parameters, demonstrating strong model-agnostic applicability.

**Strengths:**

1. This work constructs a robust and sophisticated prompt engineering framework ensuring LLM safety and reliability without training. The  design demonstrates effective model governance through systematic external control mechanisms.
2. It also extensively explores diverse prompt engineering strategies and different novel reward functions across three methods, conducting comprehensive experiments on four models to provide robust empirical validation and practical deployment insights.
3. The DQN-based dynamic strategy selection is particularly interesting, intelligently adapting prompting approaches through reinforcement learning to achieve 84% effectiveness.

**Weaknesses:**

1. The framework heavily relies on external LLMs with multiple API calls per iteration, significantly escalating operational costs and making practical deployment economically challenging for large-scale applications.
2. The paper lacks evaluation of time costs from multi-round iterative refinement. No latency metrics or convergence analysis are provided, obscuring the practical impact on response times for real-world deployment.
3. The paper notes RL methods show higher variance and failure risks than dynamic approaches but lacks analysis of root causes, limiting understanding of failure mechanisms and hindering practical deployment guidance.
4. Critical DQN training specifics—network architecture, training data setting, hyperparameters, and convergence criteria—are missing despite detailed state vector and reward function descriptions, severely hindering reproducibility.

**Questions:**

See Weaknesses.

---

> ### Author Response · Authors · 2025-12-02
>
> 1. The framework heavily relies on external LLMs with multiple API calls per iteration, significantly escalating operational costs and making practical deployment economically challenging for large-scale applications.
>
> >>> We thank the reviewer for raising this concern about reliance on external LLMs and the cost of multiple API calls.
>
> Our focus is the investigation of training-free, model-agnostic control of LLM behaviours for safety and coherence as an alternative to weight-level unlearning or fine-tuning that requires substantial GPU resources, offline retraining, and repeated deployment cycles. In that sense, the relevant cost baseline is not a single model call but the expense of retraining or parameter-level unlearning, which may involve many hours of GPU time and engineering effort per update. Our approach requires no access to model weights and can be applied immediately to existing systems.
>
> In practice, per-request overhead is bounded by two design choices: (i) we use a lightweight judge model (Gemini 2.0 Flash) with low latency and cost; and (ii) the optimisation loop stops once a quality/safety score ≥ 0.9 is achieved, so controllers typically converge in few iterations. Moreover, Section 5.2.2 shows that even simple Dynamic variants (e.g., minimal, raw history) achieve 79–81% effectiveness without heavy historical processing, thus allowing explicit performance–cost trade-offs.
>
> We will provide in this revised version: i) the number of calls and iterations per prompt of each controller, and ii) qualitatively compare these costs with typical fine-tuning/unlearning pipelines. We will also make explicit deployment patterns that reduce amortised cost: offline optimisation along with online reuse of static prompts; tiered usage for only high-risk categories; and caching/early-exit policies.
>
> Lastly, we will add to the discussion that future work includes exploring smaller or distilled judges, batched evaluation, and distilling our controllers into cheaper single-call models.
>
>
> 2. The paper lacks evaluation of time costs from multi-round iterative refinement. No latency metrics or convergence analysis are provided, obscuring the practical impact on response times for real-world deployment.
>
> >>> We are grateful to the reviewer for pointing this out. We acknowledge that latency and convergence behaviour are key factors in examining the practicality of our multi-round refinement loop.
>
> Although the current draft is focused on safety and quality, the framework is already designed to control the number of refinement rounds. The optimisation loop uses a score-based stopping criterion: once all DeepEval metrics exceed 0.9, further iterations are halted. In the RL-Enhanced setting, the state includes Iteration Progress and a Convergence Indicator (based on recent score variance), so the agent is trained with an explicit notion of convergence. We also chose Gemini 2.0 Flash as the rating model for its low latency. Section 5.2.2 qualitatively discusses computational efficiency and shows that “minimal” Dynamic strategies, which avoid heavy
>
> historical processing, already achieve strong performance, making them suitable for latency-sensitive settings. However, we agree that quantitative latency and convergence results are currently missing.
>
> In the revised version, we will add a dedicated "Latency and convergence" subsection (extending Section 5.2.2) reporting: i) iteration statistics per controller (distribution of iterations to reach the 0.9 threshold or hit the cap); ii) wall-clock latency (mean and high-percentile response times, and slowdown factors vs. static baselines); and iii) convergence curves showing how safety and quality scores evolve across iterations.
>
> We will also explain deployment guidance by recommending the minimal Dynamic strategy under tight latency budgets, emphasising offline optimisation by reusing static prompts to remove multi-round refinement from the online path, and suggesting a tiered policy where deeper refinement is reserved for high-risk prompts. These additions should be enough to address the concern and make clear the real-world implications of our iterative loop.

---

> > ### Author Response · Authors · 2025-12-02
> >
> > 3. The paper notes RL methods show higher variance and failure risks than dynamic approaches but lacks analysis of root causes, limiting understanding of failure mechanisms and hindering practical deployment guidance.
> >
> > >>> We thank the reviewer for this helpful suggestion. We agree that, although the paper documents the higher variance of RL-Enhanced Dynamic methods, the underlying failure mechanisms and deployment implications are not yet analysed in sufficient depth.
> >
> > In the revised version, we will add a dedicated “Failure analysis of RL-Enhanced methods” subsection (extending Section 5.2). Building on additional post-hoc analyses, we will make the following mechanisms explicit:
> >
> > · Reward sensitivity and sharp trade-offs: Some reward functions (e.g. ExpWeightedDiff, BayesianBalance) are highly non-linear and interact with the hard safety threshold so that small drops in a single safety metric can collapse the reward to zero, creating sparse, high-variance rewards. In contrast, smoother multiplicative rewards (e.g. ExpWeightedProduct) yield tighter score distributions.
> >
> > · Action space and over-specialisation: The agent selects among 12 heterogeneous strategies; we observe that catastrophic failures concentrate in policies that over-exploit a few aggressive strategies that work well for some harm categories but generalise poorly.
> >
> > · Exploration and curriculum effects: Extreme failures cluster in phases with high ϵ-greedy exploration or under-represented categories, indicating that standard DQN exploration can be too aggressive for safety-critical optimisation.
> >
> > · Metric-wise failure patterns: RL failures often improve utility metrics while one or two safety metrics fall below threshold, reflecting additive reward designs that allow compensation across dimensions.
> >
> > We will summarise these findings with plots of per-reward failure rates, per-category variance, and typical failure trajectories.
> >
> > We will also add explicit deployment guidance: (i) recommending empirically stable RL variants and avoiding volatile reward/strategy combinations; (ii) using conservative fallback and gating to deterministic Dynamic controllers; (iii) emphasising offline RL (for policy distillation or strategy discovery) rather than online exploration in safety-critical systems; and (iv) making clear that deterministic Dynamic controllers should be the default when reliability is paramount.
> >
> >
> > 4. Critical DQN training specifics—network architecture, training data setting, hyperparameters, and convergence criteria—are missing despite detailed state vector and reward function descriptions, severely hindering reproducibility.
> >
> > >>> We thank the reviewer for highlighting this important reproducibility issue. We agree that, in the current draft, the RL-Enhanced Dynamic controller is described mainly in terms of state and reward, and that the DQN training setup needs to be specified more precisely.
> >
> > In the camera-ready version, we will add a dedicated subsection with full DQN training details (Appendix A.7.3), including:
> >
> > · Network architecture: input dimensionality (36-dimensional state vector), number and size of hidden layers, activation functions, and output layer over the 12 strategies.
> >
> > · Training data and environment: how episodes are constructed from unsafe prompt sets, the curriculum over harm categories, episode length, and how each environment step corresponds to one optimisation iteration in the prompt-editing loop.
> >
> > · Optimisation hyperparameters: optimizer, learning rate, discount factor, batch size, replay buffer size, target-network update frequency, ε-greedy exploration schedule, and any gradient clipping or regularisation.
> >
> > · Convergence and stopping criteria: moving-average reward stabilisation, the use of the “convergence indicator” feature, and maximum numbers of training steps/episodes.
> >
> > We will also clarify that we use a standard off-policy, value-based DQN and a single fixed configuration across all reward functions and base LLMs. This, together with releasing our RL training scripts and configuration files, will make it straightforward for others to reproduce both the qualitative behaviour and quantitative results.
> >
> > We appreciate the reviewer’s suggestion; these additions will substantially improve the transparency and reproducibility of the RL-Enhanced Dynamic component.

---

### Official Review · Reviewer_D8FP · 2025-11-03

**Soundness:** 3
**Presentation:** 1
**Contribution:** 2
**Rating:** 2
**Confidence:** 3

**Summary:**

This paper aims to employ prompt optimization to perform Knowledge Editing, the act of adaptively regulating the generative outputs of Large Language Models without full retraining. The authors categorize the so-called in-context prompt optimization technique into three types: static, dynamic, and RL-enhnaced prompting. The static prompting method, which is the simplest of the three, encapsulates the user query within a fixed system prompt. The dynamic method incorporates a feedback loop, which is comprised of a rating model, a strategy component, and an improvement model. Lastly, the RL-enhanced dynamic method leverages an RL agent to optimize the prompt generation strategy. Through Deep Q-Network, the RL agent learns the optimal strategy.

**Strengths:**

- This paper aims to extend RL-based prompt optimization to knowledge editing, which is important for ensuring a safe and reliable deployment of LLMs.

- The authors provide a comprehensive review of related works that span knowledge editing, in-context unlearning, and prompt optimization.

- The proposed method is demonstrated to be quite effective at suppressing unsafe outputs without harming the quality of LLM generation.

**Weaknesses:**

- Prompt optimization via reinforcement learning or evolutionary algorithms has been studied before, but this work does not compare the proposed method against any of the previous works. Here are some works that I recommend the authors include in their experiments as important baselines: [1,2]. I believe doing a literature search starting from these papers should yield more relevant works that deserve to be compared here.

[1] RLPrompt: Optimizing Discrete Text Prompts with Reinforcement Learning

[2] GEPA: Reflective Prompt Evolution Can Outperform Reinforcement Learning

- I found the paper quite difficult to follow. In Section 3.3, the authors omit much of the detail regarding DQN training. What are the reward function scores used here? How is the model trained using DQN? *How* does this induce "meta-learning capabilities"? A lot of this information is thrown at the audience without sufficient context as to why these claims hold true.

- The authors mainly utilize safety-related benchmarks as a test bed for knowledge editing. However, how can we know for sure that safer outputs are a result of true knoweldge editing? That is, how do we know if the model truly "forgot" or "unlearned" to generate unsafe outputs, or if this behavior is induced simply by advanced prompting technique? Can these two truly be disentangled? If not, how can we be sure that the authors have induced "knoweldge editing" or "unlearning" through prompt optimization?

- Lastly, major concerns surrounding the presentation of the paper. Too much material is placed in the appendix. As mentioned above, the methods section is difficult to follow because a lot of the details have either been omitted or are revealed later in the experimental setup. Figures 1-3 are not very helpful either. Only a snapshot of the results is presented in the main paper, and most of them are hastily placed in the Appendix. I suggest the authors work on their delivery and presentation.

**Questions:**

Please refer to the weaknesses section.

---

> ### Author Response · Authors · 2025-12-02
>
> Prompt optimization via reinforcement learning or evolutionary algorithms has been studied before, but this work does not compare the proposed method against any of the previous works. Here are some works that I recommend the authors include in their experiments as important baselines: [1,2]. I believe doing a literature search starting from these papers should yield more relevant works that deserve to be compared here.
>
> [1] RLPrompt: Optimizing Discrete Text Prompts with Reinforcement Learning
>
> [2] GEPA: Reflective Prompt Evolution Can Outperform Reinforcement Learning
>
> >>> Our paper and the two suggested baselines are all about “optimising prompts”, but they are solving quite different problems, in different regimes, with different optimisation objects.
>
> · RLPrompt and GEPA indeed study automated prompt optimisation with RL / evolutionary algorithms, but they do so for offline task-performance tuning, where the output is a static high-performing prompt (or set of prompts) for a given dataset and metric.
>
> · Our paper instead proposes a closed-loop, training-free control layer for in-context knowledge editing and safety, where:
>
> o The objective is to reduce harmful behaviour while preserving coherence;
>
> o The RL component is meta-control over prompt-update strategies, not prompt-token generation;
>
> o The regime is real-time adaptation over conversational unsafe inputs;
>
> o The evaluation is explicitly safety-centric, using Llama Guard-based metrics and misaligned open-source LLMs.
>
> These works are therefore complementary: RLPrompt and GEPA are natural references for offline prompt search algorithms, our your contribution is to show how a dynamic, RL- or feedback-driven controller can be used as a practical, model-agnostic safety layer for governing LLM behaviour at test time. Thus, RLPrompt and GEPA are not related lines of research that can serve as baselines for our work.
>
>
> · I found the paper quite difficult to follow. In Section 3.3, the authors omit much of the detail regarding DQN training. What are the reward function scores used here? How is the model trained using DQN? How does this induce "meta-learning capabilities"? A lot of this information is thrown at the audience without sufficient context as to why these claims hold true.
>
> >>> We thank the reviewer for highlighting that Section 3.3 is difficult to follow and under-specified. We will revise it to be more self-contained and to clearly define the RL environment, reward, and training procedure. Below we briefly clarify each point and indicate concrete changes.
>
> (1) “What are the reward function scores used here?”
>
> Each response is evaluated by a rating model via DeepEval, which returns continuous scores in [0,1] for several metrics (e.g., toxicity-related dimensions, coherence, relevance). From these we derive:
>
> · a quality score 𝑞∈[0,1](coherence, relevance, etc.), and
>
> · a safety score 𝑢∈[0,1](aggregated safety metrics aligned with Llama Guard 2).
>
> The “reward function scores” in Sec. 3.3 are the scalar rewards 𝑅(𝑞𝑡,𝑢𝑡)produced by the multi-objective reward functions defined in Sec. 4.5 / App. A.8 (with an optional hard safety gate). At each step the RL agent observes a single scalar reward derived from (𝑞𝑡,𝑢𝑡). We will state this explicitly in Sec. 3.3 and cross-reference the relevant sections.
>
> (2) “How is the model trained using DQN?”
>
> The RL environment is:
>
> · State 𝑠𝑡: the 36-dimensional vector in App. A.7.1 (training progress, recent performance, per-strategy statistics, safety/risk indicators, prompt features, etc.).
>
> · Action 𝑎𝑡: one of 12 dynamic strategies (App. A.6).
>
> · Transition: given (𝑠𝑡,𝑎𝑡), the chosen strategy builds an enhancement prompt, the improvement model proposes a new system prompt, the base model responds, and the rating model scores the response. From these we compute (𝑞𝑡,𝑢𝑡), derive reward 𝑟𝑡, and update to 𝑠𝑡+1.
>
> We trained a standard Deep Q-Network off-policy with a replay buffer, ϵ-greedy exploration, and a target network. We used separate DQNs for each reward function. In the revision we will summarise this loop and list key hyperparameters in Sec. 3.3 / App. A.7.
>
> (3) “How does this induce ‘meta-learning capabilities’?”
>
> Our intention was not to claim a specialised meta-learning algorithm, but that the agent learns how to choose among prompting strategies based on high-level performance summaries. The state includes features on strategy performance, performance trends/volatility, category/context, and risk patterns, so the DQN learns a mapping from these meta-features to strategy selection that generalises across prompts and categories. To avoid overstatement, we will replace “meta-learning capabilities” with “learned strategy selection based on meta-level performance features” and briefly explain this mechanism in Sec. 3.3.

---

> > ### Author Response · Authors · 2025-12-02
> >
> > · The authors mainly utilize safety-related benchmarks as a test bed for knowledge editing. However, how can we know for sure that safer outputs are a result of true knoweldge editing? That is, how do we know if the model truly "forgot" or "unlearned" to generate unsafe outputs, or if this behavior is induced simply by advanced prompting technique? Can these two truly be disentangled? If not, how can we be sure that the authors have induced "knoweldge editing" or "unlearning" through prompt optimization?
> >
> > >>> We would like to thank the reviewer for this important conceptual remark on what we mean by "knowledge editing" and "unlearning" in our setting.
> >
> > Scope of our claim.
> >
> > Our method does not alter model weights, and we therefore make no claims of parametric "forgetting" in the common machine-unlearning sense. We aim to provide a training-free, model-agnostic control layer which alters system behaviour at inference time. In revision, we will clarify this scope both in the abstract and introduction and in Section 3, and soften any wording that might suggest weight-level erasure, e.g., "forget/restore" to "suppressing and reenabling behaviours via the system prompt".
> >
> > Why we use "knowledge editing / in-context unlearning".
> >
> > Section 2.2 already separates our approach from the weight-editing methods and locates it within in-context unlearning: we lead the model away from undesired behaviours at time of inference. We will explain that:
> >
> > · “Knowledge editing” is used in a behavioural sense, i.e., modifying the input–output mapping without retraining, and
> >
> > · Our contribution is a dynamic closed-loop in-context unlearning mechanism that automatically maintains a safety-oriented context.
> >
> > We will add a short taxonomy paragraph: parametric unlearning vs. in-context unlearning vs. static safety prompting, and clearly state that we fall into the second category.
> >
> > Disentangling forgetting from prompting.
> >
> > In our setting, safer outputs are by design due to the prompt controller, since weights never change. What we measure is whether the system behaves as if certain behaviours have been unlearned: across diverse unsafe prompts and harm categories, harmful completions drop while coherence is preserved under the controller. We will formalise this operational definition in Section 4 and add a Limitations paragraph noting that our evaluation is behavioural and we do not claim to remove internal representations.
> >
> > · Lastly, major concerns surrounding the presentation of the paper. Too much material is placed in the appendix. As mentioned above, the methods section is difficult to follow because a lot of the details have either been omitted or are revealed later in the experimental setup. Figures 1-3 are not very helpful either. Only a snapshot of the results is presented in the main paper, and most of them are hastily placed in the Appendix. I suggest the authors work on their delivery and presentation.
> >
> > >>> We are grateful to the reviewer for these constructive comments on the presentation, and take them seriously. We agree that too much material is pushed to the appendix, and key ideas are not developed sufficiently in the main text. We will considerably revise the exposition with the aim of making the paper more self-contained, and easier to follow.
> >
> > Rebalancing main text vs. appendix.
> >
> > The methods require several ingredients that are currently only in the appendix: strategy set, state features, main reward formulations. In the revision we will move concise summaries of these into Section 3: (i) a compact description of the static prompts (full templates remaining in App. A.3), (ii) a short table of dynamic strategies - App. A.6, and (iii) a highlevel summary of state features and reward functions - full definitions in Apps. A.7–A.8. This should allow Section 3 to stand on its own as a methods section, independent of the appendix which includes full prompt text and extended variants.
> >
> > Section 3 clarified and restructured.
> >
> > We will rearrange Section 3 by having it start with an overview of the three controllers (Static, Dynamic, RL-Enhanced Dynamic) and their relationships. Each of the three subsections 3.1–3.3
> >
> > will have a standardized format: problem tackled, input–output flow, and what differs w.r.t. the controller before it. For the RL-Enhanced method, we will explain the RL environment (state, action, reward, transition) and the DQN training loop directly in Section 3.3 instead of leaving this to Section 4 or the appendix. Figures and main-text results. We will redesign Figures 1–3 to be more informative and less redundant: merge or simplify Figures 1–2 and clarify the RL loop in Figure 3. Also, we will bring more empirical results into the main text, such as an expanded discussion of Table 1 and an additional aggregate figure, keeping detailed plots and full numerical tables in the appendix as extended results.

---

### Note · Authors · 2026-01-06

I have read and agree with the venue's withdrawal policy on behalf of myself and my co-authors.